# NIR Spectral Inversion of Soil Physicochemical Properties in Tea Plantations under Different Particle Size States

**DOI:** 10.3390/s23229107

**Published:** 2023-11-10

**Authors:** Qinghai He, Haowen Zhang, Tianhua Li, Xiaojia Zhang, Xiaoli Li, Chunwang Dong

**Affiliations:** 1College of Biosystems Engineering and Food Science, Zhejiang University, Hangzhou 310008, China; 11913056@zju.edu.cn (Q.H.); xiaolili@zju.edu.cn (X.L.); 2Shandong Academy of Agricultural Machinery Science, Jinan 250100, China; 2022120748@sdau.edu.cn; 3Tea Research Institute, Shandong Academy of Agricultural Sciences, Jinan 250100, China; zhangxiaojia@saas.ac.cn; 4College of Mechanical and Electrical Engineering, Shandong Agricultural University, Tai’an 271000, China; lth5460@sdau.edu.cn

**Keywords:** NIR spectroscopy, soil, dimensionality reduction, support vector regression, tea garden

## Abstract

Soil fertility is vital for the growth of tea plants. The physicochemical properties of soil play a key role in the evaluation of soil fertility. Thus, realizing the rapid and accurate detection of soil physicochemical properties is of great significance for promoting the development of precision agriculture in tea plantations. In recent years, spectral data have become an important tool for the non-destructive testing of soil physicochemical properties. In this study, a support vector regression (SVR) model was constructed to model the hydrolyzed nitrogen, available potassium, and effective phosphorus in tea plantation soils of different grain sizes. Then, the successful projections algorithm (SPA) and least-angle regression (LAR) and bootstrapping soft shrinkage (BOSS) variable importance screening methods were used to optimize the variables in the soil physicochemical properties. The findings demonstrated that soil particle sizes of 0.25–0.5 mm produced the best predictions for all three physicochemical properties. After further using the dimensionality reduction approach, the LAR algorithm (R^2^_C_ = 0.979, R^2^_P_ = 0.976, RPD = 6.613) performed optimally in the prediction model for hydrolytic nitrogen at a soil particle size of 0.25~0.5. The models using data dimensionality reduction and those that used the BOSS method to estimate available potassium (R^2^_C_ = 0.977, R^2^_P_ = 0.981, RPD = 7.222) and effective phosphorus (R^2^_C_ = 0.969, R^2^_P_ = 0.964, RPD = 5.163) had the best accuracy. In order to offer a reference for the accurate detection of soil physicochemical properties in tea plantations, this study investigated the modeling effect of each physicochemical property under various soil particle sizes and integrated the regression model with various downscaling strategies.

## 1. Introduction

Worldwide, the tea plant is one of the most important nonalcoholic beverage crops [1]. Tea is a perennial shrub that is mainly distributed in tropical and subtropical developing countries such as China and India. Compared with cereal crops, tea is an evergreen plant, the new buds and leaves of which are constantly being planted [2]. At 3.37 million hectares, China accounts for 63.4% of all tea harvests in the world [3]. However, the improvement in tea yield and quality cannot be separated from the impact of soil fertility.

Soil physicochemical properties refer to various natural factors in the soil, which play an important role in soil fertility. Realizing the accurate and rapid detection of the physicochemical properties of soil is beneficial for increasing tea yield and improving tea quality. At present, some analytical techniques for the detection of soil physicochemical properties have drawbacks, such as being time-consuming and involving destructive sampling [4,5].

Near-infrared spectroscopy is a commonly used non-destructive testing method, which can realize the rapid non-destructive testing of samples with high detection efficiency and low cost [6]. It is widely used in the fields of agriculture [7], forestry [8], bioscience [9], and medicine [10], among others. In recent years, a large number of scholars have constructed spectral prediction models based on the physical and chemical properties of soil. Tan [11] presented a new method for the determination of nitrogen in soil by using a near-infrared spectrum technique and random forest regression (RF), with a resulting $R_{C}^{2}$ of 0.9478 and an $R_{P}^{2}$ of 0.8743. After comparing partial least squares (PLS) with three different feature wavelength extraction methods, Ning [12] established a classification model of soil organic matter and total nitrogen content using a successive projections algorithm–extreme learning machine model. Mobasheri [13] used spectral data to model potassium in three different types of soils with an R^2^ of 0.95–0.98. Zhang [14] developed partial least squares regression (PLSR) and support vector machine (SVM) models for soil total nitrogen detection using near-infrared spectroscopy, and obtained better results. Hong et al. [15] combined near-infrared and mid-infrared spectra to predict soil organic carbon content, and the study showed that the fused data of the two spectra have great predictive potential. However, recent studies have focused on the inversion of single soil elements and little on tea plantation soil. Moreover, modeling only for a single element cannot fully predict soil fertility.

In this study, three soil physicochemical properties were modeled using a support vector regression (SVR) algorithm [16] under different soil grain size conditions to improve the prediction of soil fertility in tea plantations. Based on the successful projections algorithm (SPA) and the least angle regression (LAR) and bootstrapping soft shrinkage (BOSS) algorithms [17,18,19], the dimensionality of spectral data is reduced, and the redundancy of the sensitive band is eliminated. The specific objectives were as follows: (i) to compare which soil grain size has the best modeling results and (ii) to compare and determine which dimensionality reduction algorithm yields the highest accuracy when used for modeling.

## 2. Materials and Methods

### 2.1. Soil Sample Collection

As shown in Figure 1, soil samples for this study were collected from tea plantations in three different regions of Shandong, China, namely, Zubian Town, Junan County, in Linyi (35°10′31″ N, 118°49′56″ E); Jufeng Town, Lanshan District, in Rizhao (35°7′19″ N, 119°19′7″ E); and Wangzhuang Town, Laoshan District, in Qingdao (36°6′25″ N, 120°28′9″ E). Soil samples were collected from tea gardens in the three places that were chosen because tea gardens will be highlighted in these areas in the future. The Laoshan Mountain Range in Qingdao and the Zhushan Mountain Range in Zhushan, China, are favorable regions for the production of tea in the Jiangbei Tea Region. “A leaf rich source of a side of the people” is how Linyi’s Jounan white tea is referred to. And in China, Rizhao green tea is well-known. It is therefore possible to say that these three tea garden sections reflect certain regions in northern China.

### 2.2. Measurement of Soil Physical and Chemical Composition

These three sites yielded a total of 120 distinct soil samples from tea gardens. To maintain a constant weight for each soil sample, stones, plants, and animal remnants were removed from the collected soil, and the dirt was then dried. According to Chinese agricultural industry standard NY/T 1848–2010, “Determination of ammonium nitrogen, effective phosphorus, and available potassium in neutral and calcareous soils by combined leaching-colorimetric method”, the components of hydrolyzed nitrogen, available potassium, and effective phosphorus in the soil were determined. When ammonium ions in the leaching solution interacted with the NaCl reagent, a yellow material was created that was measured at 420 nm and whose color depth was proportional to the ammonium nitrogen content in the solution within a certain concentration range. Phosphate in the leach solution and acidified ammonium molybdate solution combined to form phosphomolybdenum heteropolyacid. It is reduced to a dark blue complex called phosphomolybdenum blue in the presence of stannous chloride, the color of which is proportional to the phosphorus content, and is measured at the 685 wavelength. A persistent potassium tetraphenylboron precipitate is created when potassium ions in the leaching solution interact with sodium tetraphenylboron. Within a particular concentration range, the turbidity is proportional to the potassium content of the solution and is measured at the 685 wavelength. The samples were measured using the mechanical shock technique. A 100 mL conical flask was filled with a fresh soil sample that weighed 2.5 × (1 + moisture content) g (accurate to 0.01 g). After adding 50 mL of soil-combined leaching agent and roughly 0.5 g of phosphorus-free activated carbon, the mixture was shaken at 220 r/min for 10 min at a temperature of 25 °C ± 2 °C. It was then dried and filtered so that the three physicochemical properties could be quickly determined.

### 2.3. Correlation Analysis of Physicochemical Properties

Following the determination of composition, Pearson correlation analysis was performed on the three physicochemical parameters. For the purpose of the Pearson correlation test, every piece of data needs to have a normal distribution. Rather, the normalization analysis revealed that the available measured potassium data did not follow a normal distribution. All of the data were logarithmized as a result. Following this procedure, the data were all analyzed and found to follow a normal distribution. Based on this, the three physicochemical properties of the soil were tested using Pearson two-tailed test. The logarithmic correlation coefficients for available potassium and the other two physicochemical properties were 0.726 and 0.640, respectively. The correlation coefficient for hydrolyzed nitrogen and effective phosphorus was 0.729. This suggests that the three physicochemical properties of the soil have a significant association with one another. As a result, there will be some overlap in the test findings of three physicochemical properties (Figure 2).

### 2.4. Soil Particle Size Classification

To obtain five particle size classes—greater than 1 mm, 0.5~1 mm, 0.25~0.5 mm, 0.1~0.25 mm, and less than 0.1 mm—the soil samples were then sieved with five different types of sieves. They are referred to as homogenous soil samples because soil samples bigger than 1 mm typically have particle sizes between 1 mm and 2 mm. Since soil samples smaller than 1 mm are challenging to reclassify, they are likewise regarded as homogeneous soil samples. All soil samples were divided into six classes of particle sizes, including the initial mixed particle size, as illustrated in Figure 3. Prediction models were then created for each of the six particle size types.

### 2.5. Soil Sample Spectral Acquisition

The soil spectra were gathered using the Intelligent Analysis Service Co., Ltd., Wuxi, China, IAS3100 NIR spectrometer. In order to ensure that the samples were evenly distributed across the bottom of the Petri dishes, soil samples of various particle sizes were placed in the dishes during collection. Each soil sample’s spectral data were calculated based on an average of 10 spectra. Each particle size class yielded a total of 120 spectra, with the detected spectral bands falling between 900 and 1700 nm. It was required to maintain the experimental temperature at 25 °C to prevent the impact of ambient factors on the spectrophotometer [20]. The spectral instrument used for the experiment is shown in Figure 4, along with the soil’s average spectrum as measured at different particle sizes.

### 2.6. Spectral Pretreatment

Using the original mixed grain size soil as a schematic, Figure 5 shows the before and after pretreatment comparison of soil spectra. Significant noise is present at wavelengths between 1600 and 1700 nm, as Figure 5a illustrates. To guarantee that the next test is unaffected, the wavelength between 1600 and 1700 nm is deleted. Only the spectral data in the 900~1600 nm range were retained for the next step of the modeling analysis.

Since the raw spectra contained noise information other than the sample’s own NIR spectral information, it was necessary to preprocess them. Based on this, the spectral data were pre-processed using multi-source scattering correction (MSC). One of the frequently employed algorithms for preprocessing hyperspectral data is multiple scattering correction (MSC). MSC can successfully remove spectral differences caused by various scattering levels, improving the correlation between spectra and data. The procedure replaces imperfect spectra with baseline translation and offset corrections.

### 2.7. Feature Wavelength Extraction Algorithm

In this paper, the spectrometer detected 800 wavelengths in the 900~1700 nm band. Even if the noisy 1600~1700 nm band was removed, there were still 700 wavelength points and the data volume was large. In order to further improve the detection accuracy and efficiency, three dimensionality reduction algorithms—SPA, LAR, and BOSS—were used to extract the feature bands from the original spectral data.

#### 2.7.1. Successful Projections Algorithm

SPA is a forward iterative search method that starts with one wavelength and then adds a new variable in each iteration until the number of selected variables reaches a set value N. Chen [21] used the SPA for feature selection based on the hyperspectral detection of sugar content in apples and obtained a high model accuracy. The purpose of SPA is to select the wavelength with the least redundant spectral information to solve the covariance problem.

#### 2.7.2. Least-Angle Regression Algorithm

LAR (least-angle regression) is a method of variable selection proposed by Efron in 2004. For a linear-in-the-parameters model, the regression target vector is a linear combination of several sets of regression variables multiplied by their coefficients. By selecting the eigenvectors step by step, one eigenvector is selected at a time to be used as the regression variable of the model, finally making a residual vector that has the same correlation and the maximum correlation with all regression variables.

The specific steps are as follows:

Step 1: Select the feature vector with the highest correlation with the initial residual (system response) as the regression vector and choose the appropriate regression coefficient for it. Calculate the residuals of the current identification model so that the residuals are equally correlated with this regression vector and another eigenvector with the highest correlation with the residuals.

Step 2: Select the eigenvector with the highest correlation with the residuals in the previous step as the second regression coefficient and choose the appropriate regression coefficient for it. Calculate the residuals of the current identification model so that this residual is equally correlated with all regression variables and another eigenvector that is most correlated with the residuals.

Step 3: Repeat step 2 to continue selecting the next regression variable and its parameters until there are no redundant eigenvectors or the selected model meets the desired residual requirements.

#### 2.7.3. Bootstrapping Soft Shrinkage Algorithm

The BOSS algorithm, proposed by Deng et al. in 2016 [17], is a novel variable selection method based on weighted bootstrap sampling (WBS) [22], bootstrap sampling (BSS) [23], and model population analysis (MPA) [2]. The linear model can be expressed as:(1)$$y=Xb+e=x_{1}b_{1}+x_{2}b_{2}+\cdots +x_{A}b_{A}+e$$
where *X* is the observation matrix, *y* is the response vector, and *b* is the regression coefficient vector.

In this method, BBS and WBS are used for the generation of random variable combinations, and MPA is mainly used for the analysis of each sub-model. Suppose the data matrix X is N × P, with one row including n samples and one column including P variables. Let the vector y of size n × 1 denote the measured properties of interest. The execution steps of the method are as follows:

Step 1: Generate k subsets on the variable space with BSS. In each dataset, the extraction is performed separately according to the parameters selected by the BSS. In this step, the same weights (W) are assigned to all the variables.

Step 2: Based on the first step, the K PLS sub-models are constructed. The prediction error (RMSECV) of each sub-model is calculated, and the optimal model is selected with its minimum value.

Step 3: The regression coefficients are calculated for each extracted model. All elements in the regression vector are changed to their absolute values, and each regression vector is expressed in unit length. By summing the normalized regression vectors, the new variable weights are obtained.
(2)$$w_{i}=\sum_{k=1}^{k}b_{i,k}$$
where *k* is the number of sub-models and *b_i,k_* is the absolute value of the normalized regression coefficient of the *k*th sub-model variable *i*.

Step 4: WBS is used for the generation of new subsets based on the weights of the new variables. This step ensures a higher probability of selecting variables with higher absolute values in the optimal sub-model.

Within the new subset, iterate steps 2 to 4 until the number of variables in the new subset equals 1. Based on this, a subset with the smallest RMSECV value after the iteration is selected and used as the optimal set of variables.

### 2.8. Modeling and Testing

Before building the prediction model, the NIR spectral data were divided into a calibration set (84 samples) and a prediction set (36 samples), according to the Kennard–Stone method, in a 7:3 ratio [24]. Next, the models for the prediction of soil physicochemical properties at different particle sizes were developed separately. The prediction model used the nonlinear SVR model. And for the kernel function, the more regionally focused and often employed RBF kernel function was selected. Then, the best particle size grade was selected, and the dimension reduction algorithms were used to compare the model precision under different dimension reduction algorithms. The overall modeling method is shown in Figure 6.

The evaluation metrics of the model include the coefficient of determination (R^2^) and the relative analysis error (RPD) [25]. The closer R^2^ is to 1, the better the model effect is considered to be. The model is considered to have high accuracy when RPD > 2, average accuracy when 1.4 < RPD < 2, and poor accuracy when RPD < 1.4.
(3)$$R^{2}=1-\sum_{i=1}^{n}{\left(y_{i}-{\hat{y}}_{i}\right)}^{2}/\sum_{i=1}^{n}{\left(y_{i}+{\hat{y}}_{i}\right)}^{2}$$
(4)RPD=SD/SEC
where $y_{i}$ is the standard value of a component of the *i*th sample, ${\hat{y}}_{i}$ is the predicted value of the corresponding component of the *i*th sample, ${\bar{y}}_{i}$ is the mean value of the corresponding component of the sample set, *SD* is the standard deviation of the analyzed samples, and *SEC* is the root mean square error of the analyzed samples.

## 3. Results

### 3.1. Analysis of Modeling Results for Different Soil Particle Sizes

The SVR regression prediction models for three soil physicochemical properties were created after dividing the spectral data of various soil grain sizes into training and validation sets. The modeling impact of soil physicochemical properties at various particle sizes is shown in Table 1.

The data in the table show that soils of various particle size categories had good prediction accuracy, but that soils with a particle size of 0.25 to 0.5 mm were the best at modeling. The prediction models’ R^2^_P_ for effective phosphorus, accessible potassium, and hydrolyzed nitrogen at the 0.25~0.5 mm particle size, respectively, were 0.955, 0.963, and 0.963. Figure 7 illustrates how three physicochemical properties of soils with a grain size under 0.25~0.5 mm affect modeling.

Larger-grained soil samples performed worse in all models than smaller-grained soil samples, and the modeling accuracy of 0.5~1 mm and 0.1~0.5 mm soils differed significantly. A too-small soil particle size, however, does not improve the accuracy of the prediction models for the relevant components, as evidenced by the fact that the effectiveness of the prediction models for the three components started to decline once the soil particle size was less than 0.25 mm. It is clear that the original mixed soil samples continue to produce better modeling outcomes, with the only soils with a particle size of 0.25~0.5 having a higher modeling accuracy. This is due to the fact that the spectrometer employed in this study scans the samples from bottom to top, and as small particles account for the bulk of the initial mixed soil samples and tend to settle to the bottom of the measurement vessel, they provide scans that are comparable to those of small homogenous soil samples.

### 3.2. Feature Band Screening Results

The soil with particles between 0.25 and 0.5 mm was the best-modeled particle size class, and utilizing the SPA, LAR, and BOSS algorithms, the distinctive wavelengths of the three physicochemical properties were recovered based on the soil spectra at this size.

Following SPA algorithm screening, there were 110, 118, and 113 distinctive wavelengths for hydrolyzed nitrogen, fast potash, and effective phosphorus, respectively. The three physicochemical properties were screened for using the BOSS algorithm at 54, 96, and 25 wavelengths, respectively. The LAR method, in contrast, used 119 feature wavelengths to check every component.

It was discovered that the concentration ranges of the distinctive wavelengths for the three physicochemical properties were roughly the same by counting the number of wavelengths repeatedly chosen by the three algorithms. According to Figure 8, the three physicochemical properties that were screened predominantly concentrated in the 900~1300 nm and 1500~1600 nm wavelength ranges. The absorbance of different soil spectra in this wavelength range varies widely. This indicates that spectra in this wavelength range can characterize the differences in the intrinsic composition of various types of soils. The characteristic wavelength ranges of the three physicochemical properties are similar. Because they are highly correlated in the previous correlation test. Thereby, the characteristic wavelengths of the three physicochemical properties are concentrated in the ranges of 900~1300 nm and 1500~1600 nm. One can observe that there is an absorption peak at 1400 nm in Figure 8a. Typically, this is thought to be produced by the O-H groups in the moisture. As a result, there is no connection between the distinctive wavelength in Figure 8b at about 1400 nm and the three physicochemical traits this work examines. This portion of the characteristic wavelength was eliminated in order to enhance the performance of the prediction model.

### 3.3. Analysis of Model Results of Different Dimensionality Reduction Algorithms

Following the dimensionality reduction algorithm’s wavelength screening of the spectrum data, separate inversion models of the physicochemical properties of various dimensionality reduction techniques combined with the SVR algorithm were produced. The best dimensionality reduction strategies for various physical and chemical parameters were examined by comparing the effects of each model. Table 2 displays the coefficient indicators for each model.

The SVR prediction models for the three physicochemical properties were greatly optimized after screening using the dimensionality reduction approach, and all the indices were enhanced. With an R^2^ of 0.976 and RPD of 6.613 for its prediction set, the LAR algorithm outperformed other algorithms in predicting the composition of hydrolyzed nitrogen (Figure 9a–c). The model utilizing the BOSS method produced the best results for accessible potassium (Figure 9d–f) and effective phosphorus (Figure 9g–i), with an R^2^ of 0.981 and 0.964 for the prediction set, respectively.

The quantitative analysis results of the three physicochemical properties via NIR were accurate and extremely reliable at the 0.25~0.5 particle size, as shown by the RPD values of all predictive models being over 4. In order to achieve quantitative prediction of soil physicochemical properties, the nonlinear model SVR in combination with the dimensionality reduction technique is superior to the model utilizing the SVR algorithm alone. Although soil samples of mixed particle size can be utilized directly for predicting the physicochemical properties of soil using NIR spectroscopy, predictions made using soil samples of 0.25~0.5 mm will be more precise.

## 4. Discussion

### 4.1. Possibility of Near-Infrared Spectroscopy for the Predictive Evaluation of Soil Fertility

This work used near infrared spectroscopy to perform non-destructive detection of nitrogen, phosphorus, and potassium in soil. NIR inversions were performed for all the key components that characterize soil fertility, in contrast to the many research works described in the introduction of this work. When the nitrogen level is greater than 100 mg/kg, the phosphorus content is greater than 10 mg/kg, and the potassium content is greater than 120 mg/kg, it is widely accepted that the soil has excellent fertility. The RPDs of all the models in this investigation are all greater than 2, and they all exhibit strong predictive performance. This shows that the technology used in this work may effectively identify soil nitrogen, phosphorus, and potassium without causing any damage, and that this methodology can then use these prediction values to assess the soil’s fertility.

### 4.2. Effects of Research Techniques on Findings

#### 4.2.1. Effect of Different Soil Particle Sizes on the Results of the Study

Six soil samples with various ranges of particle sizes were created in this work in order to completely examine the reaction mechanisms of intrinsic soil components on NIR spectra (see Section 2.1 for particular ranges). The results of the study show that models with particle sizes in the range of 0.25~0.5 mm have the best performance. Table 1 in Section 3.1 shows that the performance of the model improves with decreasing particle size. Once the particle size is smaller than 0.25 mm, the model’s performance begins to decline once more. Figure 10 displays 1 to 2 mm and 0.25~0.5 mm soils. The gaps and shadows between the soil particles will become wider and more numerous as the particle size increases. This could have some impact on the spectrum scanning procedure and alter the models. It is challenging to discern between the particles when the particle size is smaller than 0.25 mm as the particle size is too small. As a result, the performance of the model declines.

#### 4.2.2. Impact of Different Algorithmic Combinations on the Findings of the Study

The nonlinear SVR model was selected as the modeling approach in this study in order to thoroughly analyze the relationship between the response of soil physicochemical components and near-infrared spectra. And for integrated modeling with SVR, three downscaling methods with various features were chosen. Since the SPA counting rule method is simpler in theory and more useful, it was used to assess how well the other algorithms work. The LAR algorithm, which is better at handling high-dimensional information due to the large number of spectral wavelengths in this study, was selected as the second approach of dimensionality reduction. The more cutting-edge feature wavelength screening algorithm employed is BOSS. According to the experimental findings, the LAR and BOSS algorithms outperform the SPA. This demonstrates the usefulness of these two methods in this field of study.

#### 4.2.3. Conclusions on the Correlation of the Three Physicochemical Properties

This study connects the amounts of nitrogen, phosphorus, and potassium in the samples in Section 2.3. The findings revealed a strong correlation between the three elements. This suggests that these three physicochemical properties will be somewhat similar in some test results. In terms of distinctive wavelength distributions and modeling impacts, the three physicochemical properties examined in this research share certain commonalities. Their characteristic wavelengths are concentrated at 900–1300 nm and 1500–1600 nm. Additionally, their models perform better in the 0.25 to 0.5 mm range of soil particle sizes. This indicates that the results of this study can be verified with the correlation results of the physicochemical properties. This study used correlation analysis between physical and chemical attributes to verify the correctness of test findings to some degree. Still, there are drawbacks to the approach. Further research is required to examine the more intricate relationships between the three physical and chemical characteristics.

### 4.3. Research Applicability and Remaining Limitations

For this study, typical tea estates in Shandong Province, China, were chosen. The six classic Chinese tea varieties—green, yellow, black, white, green, and dark—are present in these tea gardens. Consequently, there is some generalizability of this work to northern China. In order to learn more about the fertility of the soil, near infrared spectroscopy (NIRS) was used to analyze the soil’s nitrogen, phosphorus, and potassium levels in the target tea plantation. Targeted water and fertilizer control may be implemented in the tea plantation based on the information from the detection process. Thus, the soil fertility can be regulated to the appropriate level.

However, the sample size of this study is still deficient. This is a result of the test not using a national soil. The climates in the north and south of China are very varied, and this may also affect the inherent makeup of the soil. It is possible that southern tea plantations will not be allowed to use the findings of the study. Future studies will use tea gardens around the country to conduct further research.

## 5. Conclusions

(1)Six different types of soils were gathered in the field and divided into six different particle sizes: 1~2 mm, 0.5~1 mm, 0.25~0.5 mm, 0.1~0.25 mm, <0.1 mm, and original soil samples. These samples were used to examine the effects of models for the prediction of soil physicochemical properties at different particle sizes. For soil samples with varying grain sizes, SVR models for the prediction of physicochemical properties were created, and the model with the best outcome was chosen. The experimental results demonstrated that soil with smaller particle sizes had a higher modeling effect, and the soil particle size of 0.25~0.5 mm resulted in the maximum prediction model accuracy for each physicochemical composition. Future research may directly prepare soils with this range of particle sizes. This will make it possible to acquire more accurate data on soil fertility, which can then be used to target the soil for fertilizer.(2)The SPA, LAR, and BOSS algorithms were utilized to further decrease the dimensionality of the spectral data of this grain size based on soils with particle sizes of 0.25~0.5 mm, and a fusion model of the reduced dimensionality method and the SVR algorithm was developed. The accuracy of the model is optimized well by each three-dimensionality reduction procedure. The SPA is outperformed by the LAR and BOSS algorithms. In order to detect the physicochemical components of soil, these two methods might be taken into consideration. These findings offer solid technical backing for soil sampling and testing in tea gardens, which can quickly and non-destructively identify the physicochemical makeup of the soil using near-infrared spectroscopy. Soil fertility was also evaluated by the detected soil nitrogen, phosphorus, and potassium content. This results in targeted fertilization for all types of tea tree irrigation requirements. As a result, tea gardens may be managed precisely.

## Figures and Tables

**Figure 1 sensors-23-09107-f001:**
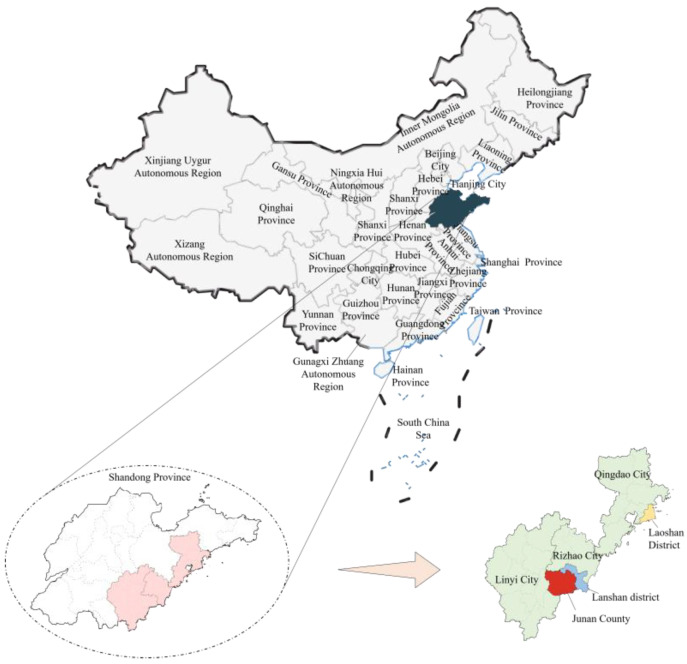
Soil sample distribution in Shandong Province, China.

**Figure 2 sensors-23-09107-f002:**
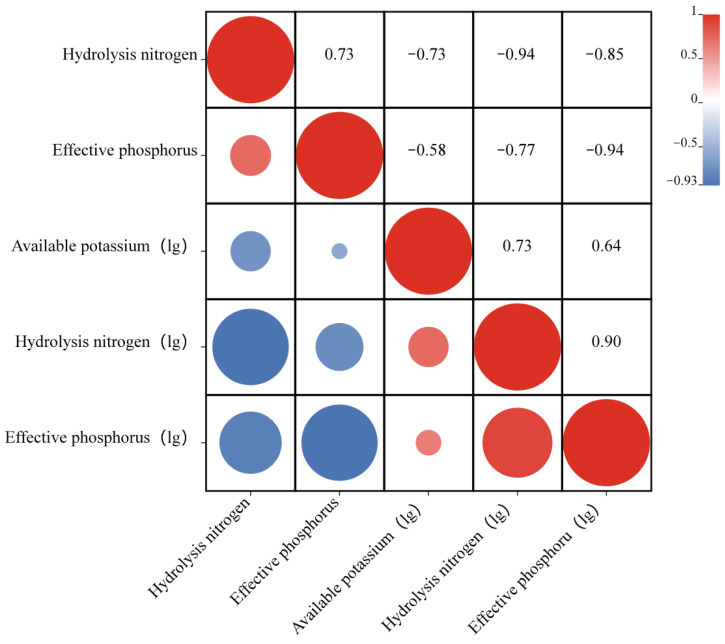
Results of correlation analysis of different physicochemical properties.

**Figure 3 sensors-23-09107-f003:**
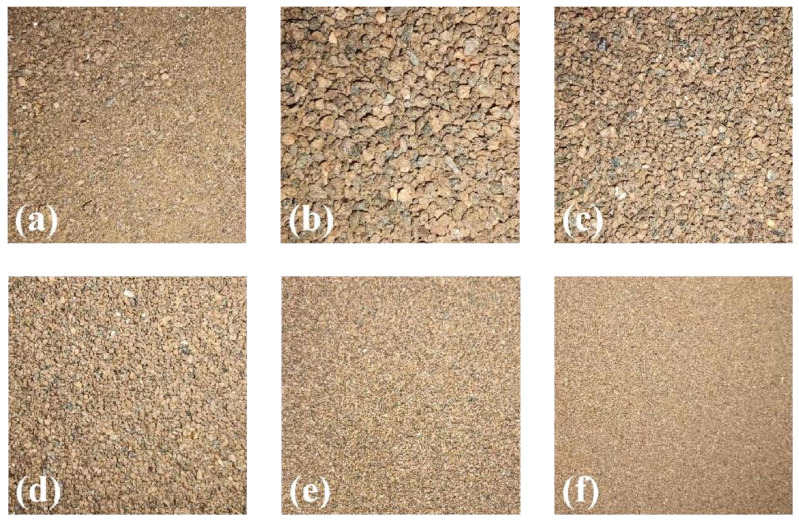
Classification of different soil particle sizes. (**a**): Original sample; (**b**): >1 mm; (**c**): 0.5~1 mm; (**d**): 0.25~0.5 mm; (**e**): 0.1~0.25 mm; (**f**): <0.1 mm.

**Figure 4 sensors-23-09107-f004:**
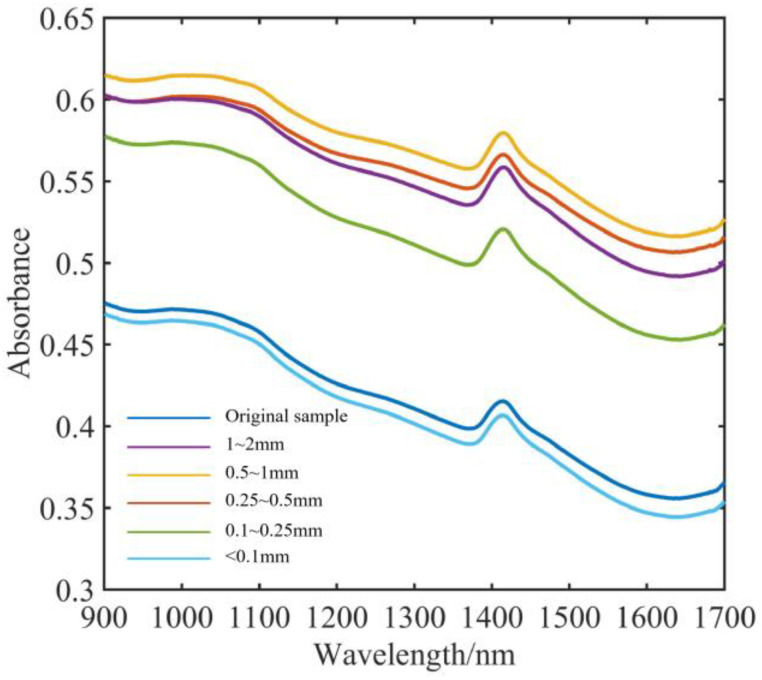
Average spectra of different particle sizes.

**Figure 5 sensors-23-09107-f005:**
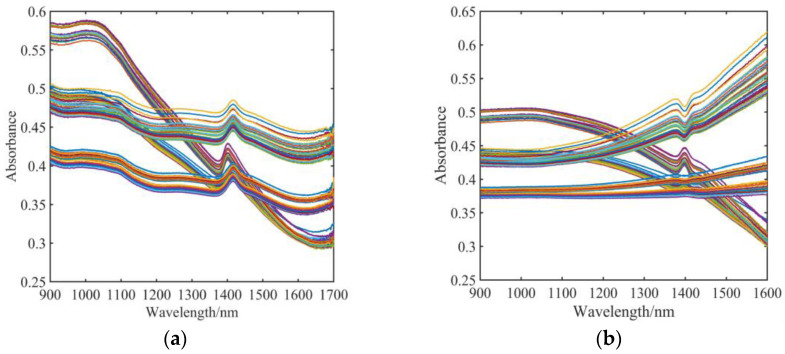
Comparison of spectral images before and after MSC processing (Shown are 150 spectra at the original particle size). (**a**): Before MSC processing; (**b**): After MSC processing.

**Figure 6 sensors-23-09107-f006:**
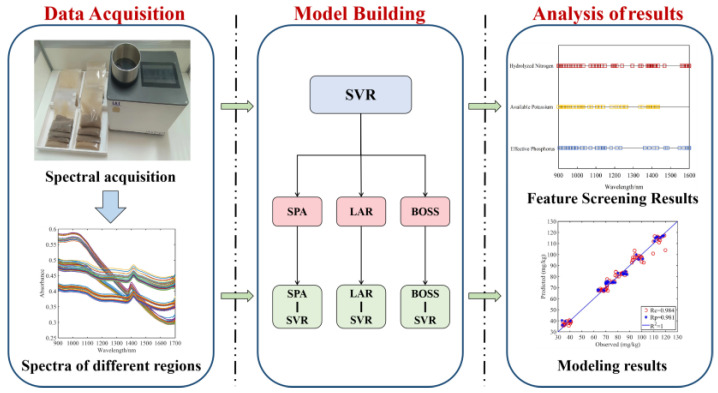
Test method.

**Figure 7 sensors-23-09107-f007:**
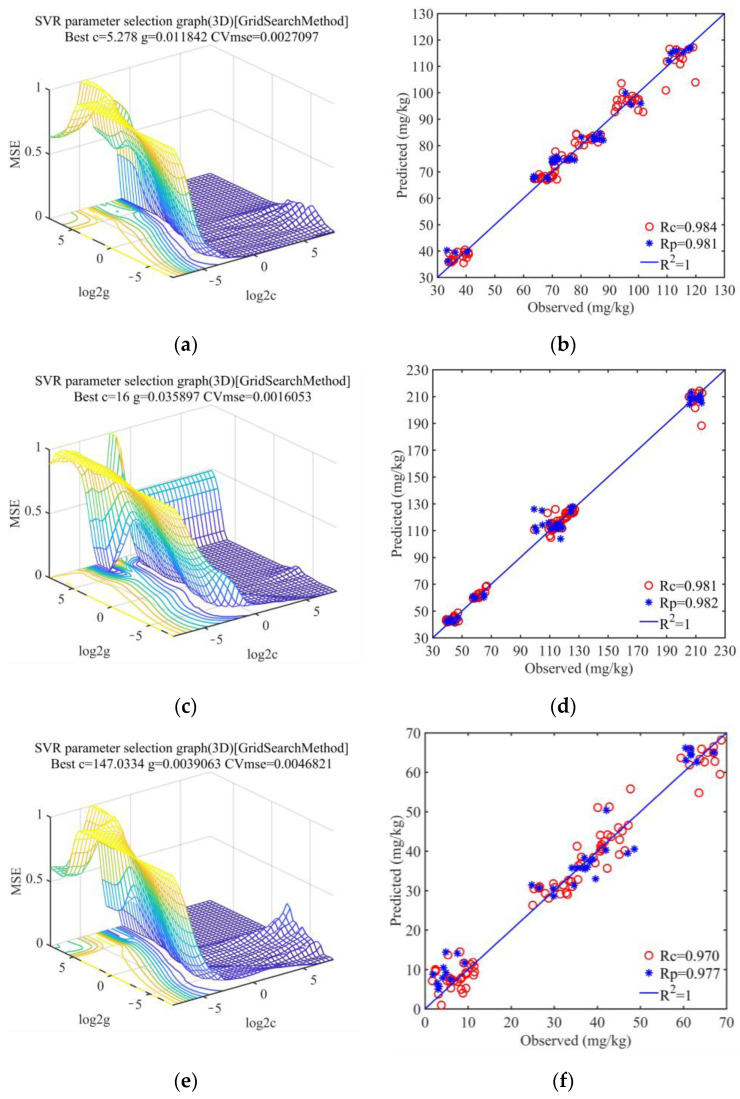
Modeling results of 0.1~0.25 mm soil particle size. (**a**) Parameter optimization of the SVR model for nitrogen hydrolysis (The darker the color of the trendline, the smaller the MSE value); (**b**) Relationship between the predicted value and the observed value set of the SVR model for hydrolyzed nitrogen; (**c**) Parameter optimization of the SVR model for available potassium (The darker the color of the trendline, the smaller the MSE value); (**d**) Relationship between the predicted value and the observed value set of the SVR model for available potassium; (**e**) Parameter optimization of the SVR model for effective phosphorus (The darker the color of the trendline, the smaller the MSE value); (**f**) Relationship between the predicted value and the observed value set of the SVR model for effective phosphorus.

**Figure 8 sensors-23-09107-f008:**
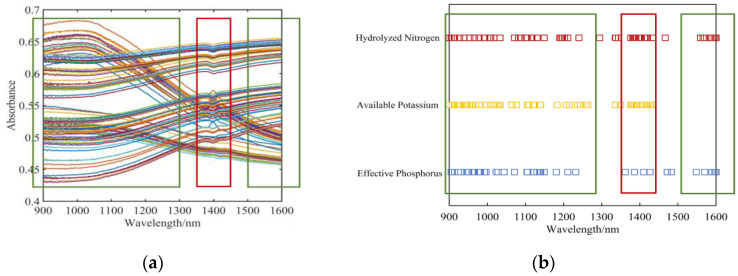
Outcomes of a distinctive band screening. (**a**) The region of concentration of the characteristic bands is at the corresponding position in the spectral curve; (**b**) Distribution of each physicochemical property’s distinctive bands.

**Figure 9 sensors-23-09107-f009:**
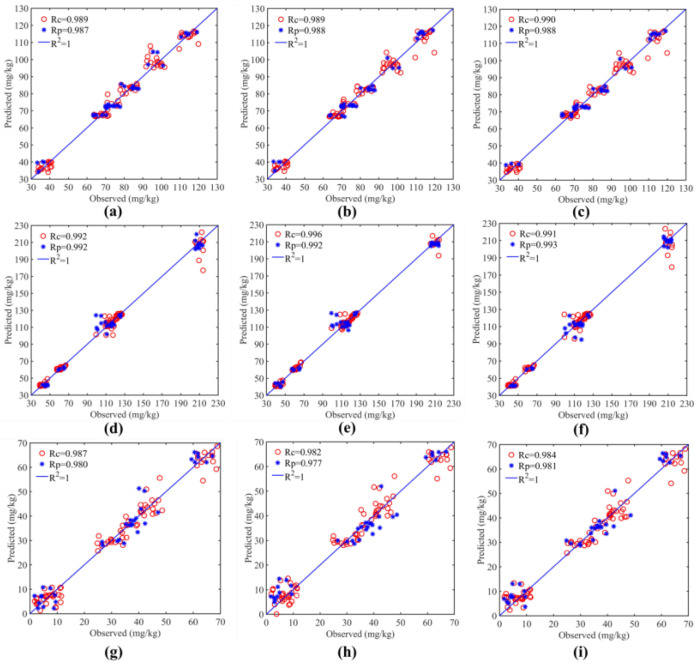
Modeling results of physicochemical properties of soil from 0.25~0.5 mm under different dimensionality reduction algorithms. (**a**) Hydrolyzed nitrogen model under SPA method; (**b**) Hydrolyzed nitrogen model under LAR method; (**c**) Hydrolyzed nitrogen model under BOSS method; (**d**) Available potassium model under SPA method; (**e**) Available potassium model under LAR method; (**f**) Available potassium model under BOSS method; (**g**) Effective phosphorus model under SPA method; (**h**) Effective phosphorus model under LAR method; (**i**) Effective phosphorus model under BOSS method.

**Figure 10 sensors-23-09107-f010:**
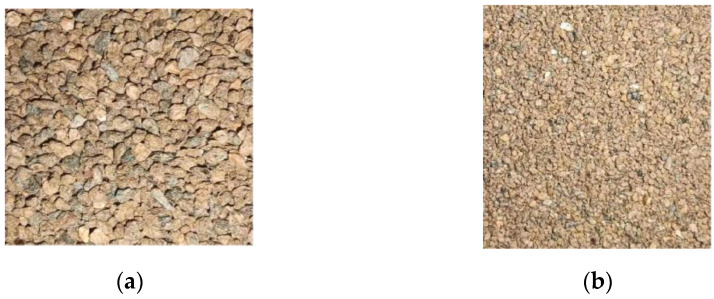
Comparison of 1~2 mm and 0.25~0.5 mm particle size soil. (**a**) Soil samples of 1~2mm particle size; (**b**) Soil samples of 0.25~0.5 mm particle size.

**Table 1 sensors-23-09107-t001:** Model results for different particle sizes of soil physicochemical properties.

Particle Size	Hydrolysis Nitrogen			Available Potassium			Effective Phosphorus		
	R^2^_C_	R^2^_P_	RPD	R^2^_C_	R^2^_P_	RPD	R^2^_C_	R^2^_P_	RPD
Original sample	0.967	0.962	4.794	0.955	0.962	4.707	0.956	0.953	4.273
1~2 mm	0.932	0.810	2.166	0.953	0.960	4.403	0.923	0.880	2.778
0.5~1 mm	0.955	0.882	2.871	0.961	0.955	4.318	0.913	0.838	2.510
0.25~0.5 mm	0.969	0.963	5.089	0.962	0.964	5.201	0.941	0.955	4.367
0.1~0.25 mm	0.976	0.945	4.199	0.968	0.962	4.963	0.966	0.952	4.028
<0.1 mm	0.962	0.881	2.851	0.962	0.959	4.938	0.932	0.869	2.693

**Table 2 sensors-23-09107-t002:** Modeling results under different dimensionality reduction algorithms.

Parameters	Modeling Methods	R^2^_C_	R^2^_P_	RPD
Hydrolyzed Nitrogen	SPA	0.979	0.975	6.249
	LAR	0.979	0.976	6.613
	BOSS	0.980	0.976	6.599
Available Potassium	SPA	0.980	0.979	6.633
	LAR	0.988	0.979	6.778
	BOSS	0.977	0.981	7.222
Effective Phosphorus	SPA	0.974	0.961	5.048
	LAR	0.964	0.955	4.660
	BOSS	0.969	0.964	5.163

## Data Availability

The data presented in this study are available on request from the corresponding author.

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
