# Peer review of "NIR Spectral Inversion of Soil Physicochemical Properties in Tea Plantations under Different Particle Size States"

_sensors, 2023, doi:10.3390/s23229107_

Round 1
Reviewer 1 Report
Comments and Suggestions for Authors
In this paper, a support vector regression model was constructed to model the hydrolyzed nitrogen, available potassium, and effective phosphorus in tea plantation soils of different grain sizes. then the successful projections algorithm and least-angle regression and bootstrapping soft shrinkage variable importance screening methods were used to optimize the variables in the soil physicochemical properties, which is useful to accutate detection of soil physicochemical features in tea plantations.
Author Response
Dear reviewer:
Thank you very much for taking the time to review this manuscript. We appreciate your comments on our research work and the content of the article. And we are honored by your recognition of our results.
Thank you for your support!
Kind regards,
Mr. Zhang

Reviewer 2 Report
Comments and Suggestions for Authors
Dear Authors
The article is very interesting; however, it lacks discussion as they describe their results. Regarding the conclusion section, although it seems there are two conclusions, it appears to be merely summarizing the results. The authors should delve into the discussion, comparing results with other studies, and presenting future lines of research to enhance their conclusions. Keywords could be shorter and start with uppercase letters. Congratulations on the work.
Beste regards
Author Response
Dear reviewer:
Thank you very much for taking the time to review this manuscript. We appreciate your comments on our research work and the content of the article. Based on your suggestions, we have made the necessary modifications. All adjustments have been accurately and successfully implemented after our best efforts. The specific response to your comment is below:
Comments 1: The authors should delve into the discussion, comparing results with other studies, and presenting future lines of research to enhance their conclusions.
Response 1: We have added a new discussion section in the fourth section of the article. In section 4.1, we also compare it to other studies; in section 4.3, we examine the remaining issues and potential avenues for further study.
Comments 2: Keywords could be shorter and start with uppercase letters.
Response 2: The keywords have been updated. They are seen in the manuscript's header.
We have also updated the article based on comments from three other reviewers, so please check it out. We would love to thank you for allowing us to resubmit a revised copy of the manuscript and we highly appreciate your time and consideration.
Kind regards,
Mr. Zhang

Reviewer 3 Report
Comments and Suggestions for Authors
1 、Please improve this part as the findings discussion is not sufficiently in-depth.
2、 The manuscript's section 2.4 contains too much repetition and too much text. Please remove it as necessary.
3 、The technique for assessing the physical and chemical characteristics of the components is described in section 2.1 of the article, but the specific stages are not given.
4、 It is advised that the conclusions of section 3.1 be more fully explained and presented.
5、Figure 9's visual information is very repetitive; please remove if necessary.
6、Please discuss in further detail how the results of the correlation analysis in section 2.1 relate to the conclusions of the thesis.
7、There should be further discussion of the limitations of the current work and a description of how it could be improved.
Comments on the Quality of English Language
Please continue to polish the English language of the document as it still has several grammatical problems.
Author Response
Dear reviewer:
Thank you very much for taking the time to review this manuscript. We appreciate your comments on our research work and the content of the article. Based on your suggestions, we have made the necessary modifications. All adjustments have been accurately and successfully implemented after our best efforts. The specific response to your comment is below:
Comments 1: Please improve this part as the findings discussion is not sufficiently in-depth.
Response 1: We have added a new discussion section in the fourth section of the article. And the findings of this paper are highlighted. We have also made some modifications in the conclusion section.
Comments 2: The manuscript's section 2.4 contains too much repetition and too much text. Please remove it as necessary.
Response 2: To chapter 2.4, several changes have been made. These modifications are emphasized.
Comments 3: The technique for assessing the physical and chemical characteristics of the components is described in section 2.1 of the article, but the specific stages are not given.
Response 3: We have added three principles of detection of physicochemical properties in Section 2.1.
Comments 4: It is advised that the conclusions of section 3.1 be more fully explained and presented.
Response 4: We have added discussion to the manuscript. The conclusion is described in detail in section 4.2.1.
Comments 5: Figure 9's visual information is very repetitive; please remove if necessary.
Response 5: Figure 9 shows the three physicochemical characteristics' model performance for various downscaling strategies. We decide to leave it alone because we think that doing so better illustrates how the models compare.
Comments 6: Please discuss in further detail how the results of the correlation analysis in section 2.1 relate to the conclusions of the thesis.
Response 6: In section 4.2.3 we describe in detail the results of the correlation analysis of the three physicochemical properties. The connection between the results and the research in this paper is also described.
Comments 7: There should be further discussion of the limitations of the current work and a description of how it could be improved.
Response 7: In section 4.3 we add the limitations of this study and future research directions.
We have also updated the article based on comments from three other reviewers, so please check it out. We would love to thank you for allowing us to resubmit a revised copy of the manuscript and we highly appreciate your time and consideration.
Kind regards,
Mr. Zhang

Reviewer 4 Report
Comments and Suggestions for Authors
Overall, the paper provides a good overview of the research conducted. However, some suggestions and comments are outlined below for further improvement:
Recommendations for Discussion
1. **Contextualize the Findings**:
- Provide a broader context for the results by discussing how they relate to existing literature and the current state of knowledge in the field. This will help readers understand the significance of your findings in the broader scientific community.
2. **Address Potential Limitations**:
- Acknowledge any limitations in the study, such as sample size, variability in field conditions, or potential sources of error in the spectral data. Discussing these limitations demonstrates a thorough understanding of the study's scope and provides a basis for future research.
3. **Compare and Contrast with Previous Studies**:
- Compare your results with findings from other relevant studies. Highlight similarities and differences and explain any discrepancies. This will help validate your findings and provide a basis for further discussion.
4. **Discuss Practical Implications**:
- Clearly articulate how the research findings can be applied in real-world scenarios.
5. **Explain the Mechanisms**:
- Provide insight into the underlying mechanisms that may be responsible for the observed relationships between particle size and the accuracy of predictions.
6. **Highlight Innovations and Contributions**:
- Clearly state the novel aspects of your approach or methodology and how they contribute to advancing knowledge in the field. This could involve discussing the unique combination of support vector regression, variable selection methods, and dimensionality reduction.
7. **Consider Alternative Explanations**:
- discuss alternative explanations for the results obtained. This demonstrates a thorough analysis and helps rule out potential confounding factors.
8. **Recommendations for Practical Application**:
- Provide specific recommendations for how the research findings can be practically applied in tea plantations.
9. **Address Future Research Directions**:
- Offer suggestions for future studies that could build upon your research. This could include investigating the applicability of the model in different agricultural contexts.
10. **Concluding Remarks**:
- Summarize the key takeaways from the discussion and reiterate the significance of your findings. Emphasize how the research contributes to advancing precision agriculture in tea plantations.
11.**Recommendations for Results and chemometric modelling**.
Provide a more comprehensive description of the chemometric methods used, including the specific steps and algorithms involved in support vector regression (SVR), successful projections algorithm (SPA), least-angle regression (LAR), and bootstrapping soft shrinkage (BOSS). This will help readers understand the technical aspects of your analysis.
**Data Preprocessing:
Explain in detail how the spectral data were preprocessed, such as baseline correction, noise reduction, and outlier removal. Justify the chosen preprocessing techniques and parameters.
**Cross-Validation and Model Selection:
Describe the cross-validation techniques used to evaluate the performance of the SVR model. Additionally, discuss how the model hyperparameters (e.g., kernel type, regularization parameters) were optimized and why those specific choices were made.
**Variable Selection Methods:
Offer a more thorough explanation of how SPA, LAR, and BOSS were applied for variable selection. Describe the criteria for selecting the most important variables and the reasoning behind choosing these specific techniques.
**Validation and Model Performance:
Present a detailed analysis of the validation results for the models, including metrics like root mean square error (RMSE), mean absolute error (MAE), and coefficient of determination (R2). Explain the practical significance of these metrics for the study.
**Interpretation of Variable Importance:
Elaborate on the interpretation of variable importance. Discuss why certain spectral features were deemed essential in predicting soil physicochemical properties and their relevance in a tea plantation context.
Sensitivity Analysis:
Conduct a sensitivity analysis to assess the robustness of the model results to variations in input data or model parameters. This can help identify potential vulnerabilities in the model's performance.
By incorporating these recommendations, it will ultimately enhance the overall quality and impact of your paper.
Comments on the Quality of English Language
English Language should be reviewed, they are some long sentences that should be shortened or explained differently.
Author Response
Dear reviewer:
Thank you very much for taking the time to review this manuscript. We appreciate your comments on our research work and the content of the article. Based on your suggestions, we have made the necessary modifications. All adjustments have been accurately and successfully implemented after our best efforts. The specific response to your comment is below:
Comments 1: Provide a broader context for the results by discussing how they relate to existing literature and the current state of knowledge in the field. This will help readers understand the significance of your findings in the broader scientific community.
Response 1: We have added a new discussion section to the article. It is added in Section 4.1.
Comments 2: Acknowledge any limitations in the study, such as sample size, variability in field conditions, or potential sources of error in the spectral data. Discussing these limitations demonstrates a thorough understanding of the study's scope and provides a basis for future research.
Response 2: In Section 4.3 we discuss the remaining limitations of the article and suggest future research directions.
Comments 3: Compare your results with findings from other relevant studies. Highlight similarities and differences and explain any discrepancies. This will help validate your findings and provide a basis for further discussion.
Response 3: In section 4.1 we discuss the differences with other studies.
Comments 4: Clearly articulate how the research findings can be applied in real-world scenarios.
Response 4: In section 4.3 we discuss the practical applicability as well as the generalizability of the results of this study.
Comments 5: Provide insight into the underlying mechanisms that may be responsible for the observed relationships between particle size and the accuracy of predictions.
Response 5: In Section 4.2.1 we discuss in detail the relationship between the response of different particle sizes and the results of the study, elucidating the principle of the mechanism.
Comments 6: Clearly state the novel aspects of your approach or methodology and how they contribute to advancing knowledge in the field. This could involve discussing the unique combination of support vector regression, variable selection methods, and dimensionality reduction.
Response 6: Our additional sections 4.1 and 4.2 describe the innovative nature of this study. In section 4.2.2 we describe the significance of the different combinations of methods.
Comments 7: Discuss alternative explanations for the results obtained. This demonstrates a thorough analysis and helps rule out potential confounding factors.
Response 7: In the discussion section, we provide a more detailed account of the findings of this paper.
Comments 8: Provide specific recommendations for how the research findings can be practically applied in tea plantations.
Response 8: In Sections 4.1 and 4.3 we describe the utility of the results in this paper.
Comments 9: Offer suggestions for future studies that could build upon your research. This could include investigating the applicability of the model in different agricultural contexts.
Response 9: In section 4.3 we present future research directions.
Comments 10: Summarize the key takeaways from the discussion and reiterate the significance of your findings. Emphasize how the research contributes to advancing precision agriculture in tea plantations.
Response 10: In the discussion section of this paper we elaborate on all the findings and explain their applicability in tea gardens.
Comments 11: Provide a more comprehensive description of the chemometric methods used, including the specific steps and algorithms involved in support vector regression (SVR), successful projections algorithm (SPA), least-angle regression (LAR), and bootstrapping soft shrinkage (BOSS). This will help readers understand the technical aspects of your analysis.
Response 11: The description of the various types of methods is described in Sections 2.4 and 2.5.
Comments 12: Explain in detail how the spectral data were preprocessed, such as baseline correction, noise reduction, and outlier removal. Justify the chosen preprocessing techniques and parameters.
Response 12: We have added about the preprocessing methods in Section 2.3.
Comments 13: Describe the cross-validation techniques used to evaluate the performance of the SVR model. Additionally, discuss how the model hyperparameters (e.g., kernel type, regularization parameters) were optimized and why those specific choices were made.
Response 13: For the SVR model we add in Section 2.5. And the significance of the use of each algorithm is described in Section 4.2.2.
Comments 14: Offer a more thorough explanation of how SPA, LAR, and BOSS were applied for variable selection. Describe the criteria for selecting the most important variables and the reasoning behind choosing these specific techniques.
Response 14: We add the mechanism of variable selection in Section 3.2. And explain the reasons for their selection in 4.2.2.
Comments 15: Present a detailed analysis of the validation results for the models, including metrics like root mean square error (RMSE), mean absolute error (MAE), and coefficient of determination (R2). Explain the practical significance of these metrics for the study.
Response 15: The methodology for evaluating the models in this paper has been described in Section 2.5.
Comments 16: Elaborate on the interpretation of variable importance. Discuss why certain spectral features were deemed essential in predicting soil physicochemical properties and their relevance in a tea plantation context.
Response 16: We add about the mechanism of selection of the feature spectra in Section 3.2.
Comments 17: Conduct a sensitivity analysis to assess the robustness of the model results to variations in input data or model parameters. This can help identify potential vulnerabilities in the model's performance.
Response 17: In Section 4.2.2 we describe the validation methods for model performance.
Response to Comments on the Quality of English Language:
We have had the article touched up by a professional agency. Please let us know if there are any problems.
We have also updated the article based on comments from three other reviewers, so please check it out. We would love to thank you for allowing us to resubmit a revised copy of the manuscript and we highly appreciate your time and consideration.
Kind regards,
Mr. Zhang

Round 2
Reviewer 2 Report
Comments and Suggestions for Authors
Dear Authors,
Your study provides some interesting insights into the use of near-infrared spectroscopy for soil analysis but could benefit from improvements in clarity the discussion of methodological choices and limitations.
The text is generally well-structured with clear subsections.
The use of near-infrared spectroscopy for non-destructive soil analysis is a powerful and relevant method. The text briefly mentions that tea gardens in the chosen regions will be highlighted in the future but does not provide a clear rationale for why these areas were selected. More context on the significance of these regions for tea production or for the study's objectives would enhance the understanding.
The description of soil sample collection, preparation, and the determination of physicochemical properties is informative. However, it might benefit from more detailed information about the sampling techniques used and any potential sources of bias.
The use of Pearson correlation analysis to explore the relationship between physicochemical properties is a valid approach. However, the text does not discuss the assumptions or limitations of this analysis. A brief mention of why logarithmization was necessary would be helpful for readers.
The explanation of why the 1600~1700nm band was removed is missing. Providing a rationale for this preprocessing step would strengthen the methodological transparency.
Beste regards
Author Response
Response to Reviewer 2 Comments
Dear reviewer:
Thank you very much for taking the time to review this manuscript. We appreciate your comments on our research work and the content of the article. Based on your suggestions, we have made the necessary modifications. All adjustments have been accurately and successfully implemented after our best efforts. The specific response to your comment is below:
Comments 1: The text briefly mentions that tea gardens in the chosen regions will be highlighted in the future but does not provide a clear rationale for why these areas were selected. More context on the significance of these regions for tea production or for the study's objectives would enhance the understanding.
Response 1: The article's section 2.1 now accurately describes this part. It was highlighted why these three locations were selected.
Comments 2: The description of soil sample collection, preparation, and the determination of physicochemical properties is informativeHowever, it might benefit from more detailed information about the sampling techniques used and any potential sources of bias.
Response 2: We have added details on the determination of the content of physicochemical components in Section 2.2.
Comments 3: The use of Pearson correlation analysis to explore the relationship between physicochemical properties is a valid approach. However, the text does not discuss the assumptions or limitations of this analysis. A brief mention of why logarithmization was necessary would be helpful for readers.
Response 3: We updated the correlation analysis in Section 2.3 and described the limitations of the method in Section 4.2.
Comments 4: The explanation of why the 1600~1700nm band was removed is missing. Providing a rationale for this preprocessing step would strengthen the methodological transparency.
Response 4: We have added this part in section 2.6.
Thank you for reviewing this manuscript. In addition to your suggestions we have made some updates to other parts of the manuscript. Please allow us to upload the new manuscript. You can access it in the system.
Kind regards,
Mr. Zhang

Reviewer 4 Report
Comments and Suggestions for Authors
The paper was well corrected.
Comments on the Quality of English Language
no comments
Author Response
Response to Reviewer 4 Comments
Dear reviewer:
Thank you very much for taking the time to review this manuscript. We also revised the manuscript based on comments from other reviewers. Please allow us to provide a new manuscript. We appreciate your comments on our research work and the content of the article.
Kind regards,
Mr. Zhang
